# No Motor Costs of Physical Education with Eduball

**DOI:** 10.3390/ijerph192315430

**Published:** 2022-11-22

**Authors:** Ireneusz Cichy, Agnieszka Kruszwicka, Tomasz Przybyla, Weronika Rochatka, Sara Wawrzyniak, Michal Klichowski, Andrzej Rokita

**Affiliations:** 1Department of Team Sports Games, Wroclaw University of Health and Sport Sciences, Mickiewicza 58, 51-684 Wroclaw, Poland; 2Learning Laboratory, Adam Mickiewicz University, Poznan, Szamarzewskiego 89, 60-568 Poznan, Poland

**Keywords:** educational balls, graphomotor skills, gross motor, learning, locomotor skills, object control, primary education, space-time orientation

## Abstract

Numerous neuroscience studies demonstrate that when motor and cognitive tasks are performed simultaneously, there is dual-task interference. Experiments show that the cost is a temporal deterioration in motor functioning. However, there is no comprehensive research on the developmental costs of dual-task exercises incorporated into physical education (PE). Such an approach is called the interdisciplinary model of PE and is used to stimulate cognitive development. Therefore, there is a knowledge gap regarding the motor costs of methods based on this model, e.g., Eduball. The Eduball method integrates core academic subjects with PE using a set of educational balls printed with letters, numbers, and other signs. To fill this knowledge gap, we replicated the Eduball experiment, focusing on motor development. The half-year intervention occurred in one primary school class. The control group was a peer class participating in traditional PE, not based on dual tasks. We tested students’ space-time orientation and graphomotor, locomotor, and object control skills. We found no motor costs of the intervention. Eduball-based PE stimulated motor development as much as traditional PE. Our study suggests that methods based on the interdisciplinary model of PE are safe for motor development. As such, it is worth considering their use in children’s education.

## 1. Introduction

Conventionally, cognitive processes and mechanisms controlling physical activity have been considered unconnected. However, there is new evidence of cognitive-motor interference [1]. Recent cognitive neuroscience studies [2,3,4,5,6,7] have shown that brain areas of higher cognitive control are activated while performing actions previously recognized as automated. These observations indicate substantial attentional requirements for controlling such simple activities as walking or running [8]. Moreover, it was detected that motor performance decreases when a cognitive-motor task is realized [9]. For example, when somebody is running, run speed drops immediately if she or he starts doing some cognitive subtasks, such as analyzing data from a smartwatch about training effectiveness [10]. This phenomenon occurs since specific cognitive processes—e.g., attention—jointly safeguard movement [11]. Consequently, when simultaneously carrying out a cognitive and a motor task, attention resources needed to ensure safe movement are limited. In order to increase safety, the brain slows the run down, or—in the case of other activities—it reduces, for example, the precision of the movement or the power of the throwing. However, the accident risk is still high [12]. Furthermore, dual-task activity may also decrease cognitive performance, mainly if it is challenging to reduce physical activity intensity significantly [13]. This effect, i.e., deteriorations in one or both task performances that occur when the motor and cognitive tasks are simultaneously performed, is called “dual-task costs” (or “dual-task interference”) [14]. Dual-task costs increase as the cognitive task becomes more complex and/or the motor task becomes more intense [10,12].

One of the most up-to-date forms of dual-task is smartphone use while walking [13]. Both experiments and medical statistics confirm that it has high motor costs, demonstrated by frequent—and sometimes severe—falls [15]. Therefore, in the Smart Education concept, based on mobile learning [16], it is recommended that students do not perform tasks on smartphones while moving around, e.g., in the city or park [17]. Learners must separate motor and cognitive activities [18]. For example, when learning in the so-called CyberParks, they have to sit in special shelters to use smartphones. When they change places or look for something in such a park, the smartphone should be in their pocket [19]. However, there are also educational concepts where students intentionally perform dual tasks. These include various forms of dual-task exercises or training [20,21]. Such activities are the foundation of the interdisciplinary model of PE—an approach that entails incorporating core academic subjects into the PE curriculum [22,23]. Strategies based on this model bring significant cognitive effects, such as improving attention or stimulating mathematical and linguistic development [24,25,26,27,28,29,30]. An example is the Eduball method which merges movement with learning content using a set of educational balls for team mini-games printed with letters, numbers, and other signs [31]. The explanation for this phenomenon is that dual tasks activate the prefrontal cortex, which plays a role in executive functions such as attention [13]. Therefore, training involving various types of dual tasks stimulates these functions [32]. It can also be explained by an embodiment of linguistic and mathematical development [33,34]. After all, language development is firmly related to gesticulating [35] and counting development to using fingers [36]. Consequently, these two functions have common neuronal mechanisms with motor functions, e.g., with praxis [37,38,39,40,41,42]. Eventually, specific motor-cognitive training can increase young children’s communication and numerical skills [43,44,45,46,47,48,49]. In other words, although dual tasks are associated with high cognitive costs, they are natural for developing children. As such, they can be an effective tool for intellectual development, but only under controlled and deliberately designed conditions.

However, there is a knowledge gap regarding the consequences of dual tasks on motor development. This is because studies on the interdisciplinary model of PE have focused on cognitive effects and marginalized the analysis of motor development. Alternatively, this model was tested only in one motor area using one motor test [50,51,52,53,54,55,56,57,58]. No studies monitored different aspects of motor development in one experiment, although scientists pointed to such a need [27]. This also applies to experiments on the Eduball method, even when they focused on motor effects [59,60,61]. Thereby, whether or not the application of the interdisciplinary model of PE has motor costs, i.e., weakens students’ motor development, remains an open question. Nevertheless, this problem is critical. In primary schools, the time devoted to PE is still being reduced. Now it is less than 10% of the total school time [62]. Thus, PE’s fundamental goals, such as developing physical and health literacy, active lifestyles, and preventing chronic diseases and mental health disorders, are becoming more threatened [63]. Consequently, if the interdisciplinary model of PE generates motor costs, these goals will be even more challenging when teachers follow this approach. Instead of giving a chance for holistic development, the interdisciplinary model of PE will contribute to an even greater emphasis on cognitive development and prioritize core academic subjects, such as literacy and numeracy, over PE [23,64]. To address this issue and knowledge gap, we replicated the Eduball experiment and included three motor tests in the procedure. In short, the intervention consisted of various Eduball games incorporated into PE lesson plans in one primary school class for half a year. Before and after the intervention, we tested students regarding their space-time orientation, as well as graphomotor, locomotor, and object control skills. The same tests took place in the control class, which participated in the traditional PE program this semester. We hypothesized that there would be no motor costs of PE with Eduball, because, while we are adding cognitive elements to the movement, this method still emphasizes the quality of PE. More broadly, we assumed that using the interdisciplinary model of PE in children’s education poses no threat to their motor development.

## 2. Materials and Methods

### 2.1. Participants

Twenty-seven Polish students from two second-grade classes (13 girls, ages 7–8, mean = 7.52, *SD* = 0.51) participated in the study. Informed consent was obtained from their parents or legal guardians. Both classes attended the same school located in a large city in Poland. Classes were randomly assigned to control and experimental groups using Research Randomizer—a random group generator available at https://www.randomizer.org/ (accessed on 28 August 2017). The control group comprised 12 pupils (5 girls, mean age = 8.00, *SD* = 0.00), while the experimental group included 15 learners (8 girls, mean age = 7.13, *SD* = 0.35). The sample size was calculated using the standard formula in G*Power (Version 3.1.9.6) with type one error (*α*) of 0.05 and type two error (*β*) of 0.20 (1 − *β*/power = 80%). We found that a minimum of 24 participants are required, with 12 in each group (with an expected large effect size). Inclusion and exclusion criteria were the same as in our earlier studies (e.g., [27]). In short, inclusion criteria were as follows: being a learner of the selected class and frequently participating in activities during the investigational period, while exclusion criteria were as follows: contraindications for participation in PE and omitted pre-test or/and post-test. All the schoolchildren met the inclusion criteria. Nobody was excluded from the study.

### 2.2. Procedure

Our study was realized at school and lasted one semester. The procedure was approved by the local Ethics Committee for Research Involving Human Subjects and followed the principles of the Helsinki Declaration.

We used the technique of parallel groups. The experimental and control classes followed the same curriculum based on the Polish National Ministry of Education core program. In both groups, the PE teacher taught all three 45-min PE classes per week. Two of them were held in the experimental class using the Eduball method. The remaining class ensured the implementation of the formal requirements of the curriculum. As in the Eduball experiments replicated here [24,25,26,27,31,59,60,61], various activities from the Eduball games set [65,66] were integrated with PE lesson plans. The games were chosen in compliance with the curricula and the day’s theme. A brief description of the Eduball method is presented in Figure 1. For detailed specifications, see the Eduball papers [24,25,26,27,31]. All PE classes were conducted in the control group without Eduball, following the standard PE program. In other words, in the control class, the PE teacher conducted PE under the aims and objectives of the school’s program for developing physical fitness and health education.

As shown in Figure 2, the experiment involved two measurement periods: a pre-test at the beginning of the school year (September) and a post-test at the end of the first semester (January). During both stages, we diagnosed space-time orientation skills, graphomotor dexterities, and fundamental motor abilities. For this purpose, we used the Reactive Shuttle Drill test, the MovAlyzeR test of pen pressure, and the Test of Gross Motor Development (Second Edition). In the following sections, we characterize each of these tests.

#### 2.2.1. Reactive Shuttle Drill Test

We used the modified Reactive Shuttle Drill test to evaluate space-time orientation, one of the critical skills of human movements [67,68]. This test is child-adapted. As shown in Figure 3, it is about running as fast as possible through five randomly chosen paths. Every path always starts and ends on the mat called Smart JUMP Mat. The halfway point of each path is a gate. Which path to take is indicated by a light column and a light reflector placed in the targeted gate.

We ran this test via the Fusion Smart Speed system (Fusion Sport, Brisbane, QLD, Australia). As recommended by the test authors [67], we first separated the 25 × 25 m area at the sports hall. We secured this space to increase the safety of the participants: e.g., an area around the gates was protected with mattresses, and the reaction mat was fixed to the ground. The test began with the introduction part when the experimenter explained the proper execution of the task. Then, the student stood at the taped-out starting line with his/her back to the goals. This line was located one meter from the outer edge of the mat. At the start signal, the participant ran up to the mat. The timing was automatically registered by a computer. The measurement started when the child stepped on the mat, and the randomly chosen gate column light went on. Next, the student ran straight to the so-marked gate, crossing the tape-marked line with both feet and returning to the mat. This mat—goal—mat pattern was repeated five times with a different, randomly selected goal. The child had to perform the whole test twice. Each attempt lasts about 25 s. The break between them takes 15 min. The shorter the time, the better the space-time orientation. Our analyses included only the better result for a given student.

#### 2.2.2. MovAlyzeR Test of Pen Pressure

The MovAlyzeR (NeuroScript LLC, Tempe, AZ, USA) test was used to evaluate pen pressure, an essential indicator of graphomotor skills in children. MovAlyzeR captures pressure online when participants write using the Wacom Pro electronic pen (Wacom Co., Ltd., Kazo, Saitama, Japan) on the horizontally positioned 13-inch Wacom Cintiq Companion 2 graphic tablet (Wacom Co., Ltd., Kazo, Saitama, Japan) [69,70].

We conducted the test in the teacher’s office, where one experimenter and the participant were together. The chair and table were adjusted to the age of the children. The experimenter began the test by introducing its aims and completing the tutorial task. During this time, the participant was also familiarized with how to use the tablet. This training phase consisted of writing the student’s own name twice (Figure 4a). Then, eight formal trials started. Each student had to write one Polish word four times: twice on a blank screen (Figure 4b) and twice with letter placement on three lines (Figure 4c). Next, the participant had to write two pseudo-words in lines (Figure 4d,e). Students always used the dominant hand. MovAlyzeR measured pen pressure from when the child started writing until it ended, taking a sample every 10 milliseconds (100 Hz). The device computed the average pen pressure for each trial. Finally, MovAlyzeR calculated the average pen pressure for the whole test. The lower the average, the lower the pen pressure, and the lower the pen pressure, the higher the graphomotor skills. In older children who are already very skilled in writing, the average value is close to 200, but in children in early school, the scores are much higher.

#### 2.2.3. Test of Gross Motor Development—Second Edition

We ran the Test of Gross Motor Development—Second Edition (PRO-ED, Austin, TX, USA) to assess fundamental movement skills. This test—abbreviated as TGMD-2—is dedicated to children aged 3–10. It evaluates the performance of six locomotor and six object control abilities. Figure 5a–f shows that the first skill set includes running, galloping, leaping, jumping, hopping, and sliding. The second one (see Figure 5g–l) is limited to throwing, catching, kicking, striking, dribbling, and rolling.

We conducted TGMD-2 at the sports hall during a school PE session. The measurement was as follows. One experimenter demonstrated the proper execution of locomotor and object control skills. Then the children re-demonstrated these actions in the same order. The participant had to complete one practice and two formal trials. Four experimenters observed and scored each re-demonstration using criteria embedded in TGMD-2. If the child met the given criterion, a score of 1 was awarded. If not, the score was 0. There were three to five criteria for each category [71,72]. The participant could score a maximum of 24 points for the locomotor subtest and the same number for the object control subtest. The higher the score, the better the performance. We used the standard TGMD-2-kit [72], including one 9-inch playground ball, one basketball, one soccer ball, one 4-inch lightweight ball, one tennis ball, one softball, one 4.5-inch square beanbag, tape, two traffic cones, one plastic bat, and one batting tee.

### 2.3. Data Analysis

The dependent variables were space-time orientation, graphomotor, locomotor, and object control skills. All variables were expressed in the mean scores and calculated separately for the control and experimental groups and pre-test and post-test. Firstly, we applied paired samples Student’s *t*-test to compare the pre-test and post-test scores. Secondly, we calculated changes between post-intervention and baseline measurements (Δ) by the formula: post-test scores − pre-test scores. Then, we ran independent samples *t*-test to determine the significant difference between groups’ Δ. We used Welch’s *t* because the groups were unequal [73]. For all *t*-analysis, an effect size was calculated using Cohen’s *d* [74]. Finally, we run an analysis of covariance (*ANCOVA*) with learners’ pre-test scores set as the covariate and post-test scores as the dependent variable. If pre-test scores accounted for a significant proportion of variance, we calculated estimated marginal means (*EMM*) to correct the model we are testing. We used partial eta squared (*η*²*p*) in this step to measure effect size [75]. The necessary post hoc tests were performed with additional Tukey’s correction and again with Cohen’s *d*. All statistical analyses were carried out using jamovi for Mac (Version 2.3.0.0) [76,77,78,79]. The adopted level of significance was *α* = 0.05. Anonymized data supporting this study’s findings are publicly available in Open Science Framework at https://osf.io/6pfrt (accessed on 8 September 2022).

## 3. Results

Regarding space-time orientation skills, the control group did not improve its result during the experimental period. Moreover, its result worsened when comparing the pre-test with the post-test (by 215 milliseconds), but this is not a significant difference (see the top panel of Table 1). The experimental group, at the same time, improved its result by 618 milliseconds. Although the size of this effect was medium, the observed increase was significant (see the bottom panel of Table 1). The distinction in difference scores between the groups (833 milliseconds) was also significant (see Table 2). However, Table 3 shows that the pre-test result influenced this effect. After correcting the model, the difference between the groups dropped to 300 milliseconds, and with Tukey’s *p* = 0.256 became only a trend (see Figure 6a).

In the pre-test, the control group fared slightly worse than the experimental group in graphomotor skills. Their average pen pressure was higher by 87.24 points. However, as depicted in the top panel of Table 1, a paired samples comparison of the pre-test and post-test scores showed that this group significantly improved its performance throughout the study period. It reduced average pen pressure by 64.87 points. The experimental group performed the pre-test and the post-test at a similar level. We found here an insignificant difference of 3.35 points (see the bottom panel of Table 1). Nevertheless, the differences in changes in graphomotor skills between groups (68.23 points) were not significant (see Table 2 and Figure 6b), and the above-mentioned slight differences in pre-test scores did not affect our model (see Table 3).

Concerning gross motor development, we observed some regression in the control group (see the top panel of Table 1). This effect occurs mainly in object control. We found deterioration for the whole category (over 2.5 points) and for striking (1.08 points). We also detected a negative trend in dribbling and throwing (0.58 and 0.75 points, respectively). There was also a negative tendency in locomotor skills (*p* = 0.071, and the overall deterioration was slightly more than 1 point). We did not observe such effects in the experimental group (see the bottom panel of Table 1). It only deteriorated in the area of dribbling (by 1.13 points), but with the effect size below expectations. We also found here a negative trend in sliding, amounting to 0.53 points. However, this group improved their kicking score (by 0.67 points), and it was a large effect in Cohen’s terms. Comparing the changes between the groups (Table 2), we found differences in the overall object control score (2.78 points) and in the kicking range (0.75 points, see Figure 6f). However, the first category required correction (see Table 3), and this effect did not occur in the corrected model (the difference decreased to 0.80, Tukey’s *p* = 0.551, see Figure 6d). The effect size in the case of the second category was large, although the power was below expectations (1 − *β* = 56%).

## 4. Discussion

Implementing three extensive motor tests into the Eduball-experiment’s procedure, we confirmed the hypothesis that integrating PE with cognitive content does not slow down students’ physical development. In other words, we found no motor costs in the interdisciplinary model of PE. However, our investigation has some limitations, and more research is needed to fully confirm the motor safety of this approach. We also observed some trends suggesting that Eduball-based PE may give better motor profits than traditional PE. Nevertheless, the tested model’s motor benefits premise is just a hypothesis. Below we discuss all these observations in greater detail. We also speculate on what experiments should be done to overcome this project’s limitations.

### 4.1. Eduball Costs for Motor Development

The tests used in this study allowed us to analyze three primary areas of child motor development. First, we tested a person’s ability to orient himself in terms of space, time, and surroundings, i.e., space-time orientation [80]. Second, we studied graphomotor dexterities—some fine motor skills used for writing [81]. Third, we have included fundamental movement skills such as running, galloping, hopping, leaping, jumping, sliding, striking, dribbling, catching, kicking, throwing and rolling [82]. Our results demonstrated that the Eduball method does not induce regression in students’ motor development in these areas. Thus, using this method incurs no motor costs. However, it should be emphasized that we did not detect a radical improvement in physical fitness throughout a one-semester experiment in both the Eduball and no-Eduball groups. The explanation may be as follows. The critical period of motor development occurs during the early years of childhood [83,84]. Regarding fundamental motor skills, children should be competent in locomotor and object control abilities to a certain level by the age of 6 or 7 [83,84,85]. Moreover, children between the ages of 6 and 8 appear to experience less motor progress in gross and fine motor skills than children between the ages of 9 and 11 [86]. Therefore, it seems that the observed predominant lack of progress in motor development could be linked with the age of the participants, i.e., 7–8 years old. It can also be associated with motor development characteristics in this age group and relatively short-term intervention. Future research should consider this issue with a more extended influence period. A one-year experiment should improve motor skills enough to be more noticeable or detectable by motor tests and statistical methods. Investigations could also include students at every primary school level to investigate developmental trajectories in motor skills among different age groups.

### 4.2. Eduball Profits for Motor Development

In addition to our observations that the Eduball method does not adversely affect the children’s motor development, our outcomes also show that it generates some motor profits. In other words, incorporating language or mathematics content into PE stimulates motor development. We observed some positive influence of the Eduball method on gross motor development, i.e., a trend in the progress of space-time orientation abilities and a substantial effect on kicking learning, a part of object control development. Two factors may play a role here. The first one is a PE innovation. Many studies [87,88,89] show that novelties introduced to PE activate children and boost them to be thorough and involved in motor tasks. As a result, any PE innovations accelerate physical development. The second factor is the specificity of physical activities with Eduball. Most exercises in traditional PE, such as basketball, high jumps, or table tennis, are dominated by unilateral motor tasks [90]. Thus, they increase the body’s functional asymmetries [91,92]. PE with Eduball differs in this respect from conventional PE. It tends towards motor symmetry [26,27]. For example, the Eduball method forces schoolchildren to practice bilateral [93] and non-dominant hand/leg training [94,95]. As the effect of an interhemispheric transfer [96,97], Eduball may increase the efficiency of PE in some motor parameters, mostly in object control skills (for more on a bilateral transfer effect, see [98,99,100]). Studies including longer than a one-semester intervention (e.g., a one-year experiment) are, however, needed to confirm this possible beneficial effect. Also, long-term effects should be examined, for example, by organizing a follow-up test half a year after the intervention.

### 4.3. Eduball Experiment’s Limitations

One of the study’s limitations is that it covers quite a short intervention. Therefore, to better assess the impact of the interdisciplinary model of PE on motor development, future investigations should be more prolonged, e.g., two semesters. Another limitation is measuring only immediate effects. In subsequent projects, follow-up tests should also examine long-term effects. For example, an additional measurement after the holiday or the following semester can be applied. The main limitation is, however, the use of only one method (Eduball). In further studies, a different strategy based on the interdisciplinary model of PE should be involved. To eliminate the problem of the effect of innovation as such, researchers should also arrange a different control group applying non-dual-task-based innovation. We also recommend that the classes be more numerous or to have two classes per condition. This will provide better power of tests. Additionally, upcoming experiments might include several educational levels, e.g., all grades of primary school. They should be carried out in more than one country, considering different cultural traditions and approaches to physical activity.

## 5. Conclusions

The interdisciplinary model of PE is becoming more and more popular. Many studies have shown that it has many benefits for the academic development of children. However, the theory of dual-task costs suggests that linking PE with cognitive tasks, e.g., with counting, may cause deterioration in physical development. The question of whether or not dual-task-based PE has such motor costs remained unanswered. Here, we show that using the Eduball method does not slow down students’ growth in such motor areas as space-time orientation, graphomotor and fundamental movement skills. Thus, our outcomes, yet still with some limitations, suggest that methods based on the interdisciplinary model of PE are safe for physical development. More broadly, our report demonstrates that although motor-cognitive training is generally associated with dual-task costs, it may be an effective educational strategy. We, therefore, recommend using the interdisciplinary model of PE in children’s education.

## Figures and Tables

**Figure 1 ijerph-19-15430-f001:**
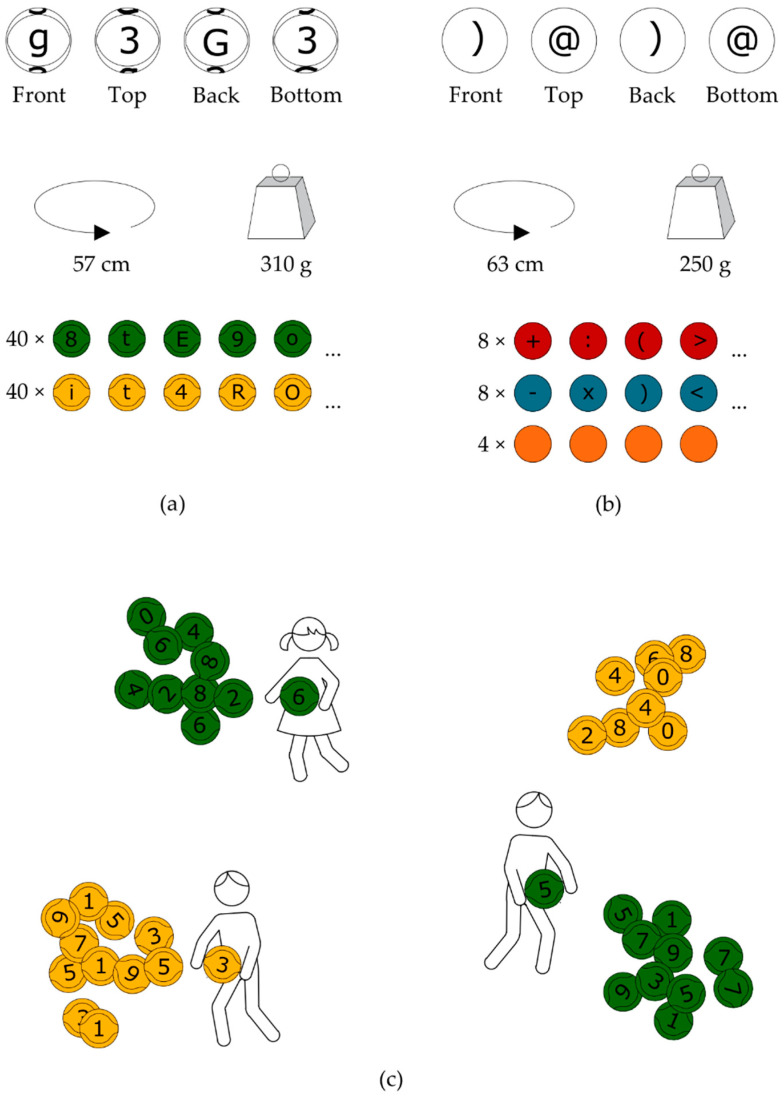
Eduball method. The Eduball set consists of 100 balls sized to fit a child’s hand, divided into two subcategories. (**a**) The first ball category includes green and yellow basketballs (size 3) with printed black numbers (from 0 to 9) and letters (uppercase and lowercase on opposite sides) of the Polish alphabet; (**b**) the second ball category contains volleyballs (size 4) in red and blue with mathematical operations, “at” (@) signs, and unprinted orange balls used as universal blanks; (**c**) an example of an Eduball game called “Even and Odd Numbers”, in which students’ task is to group the balls scattered around a sports hall into four categories as quickly as possible: even yellow, odd yellow, even green, odd green (the game aims to consolidate the recognition of even and odd numbers).

**Figure 2 ijerph-19-15430-f002:**
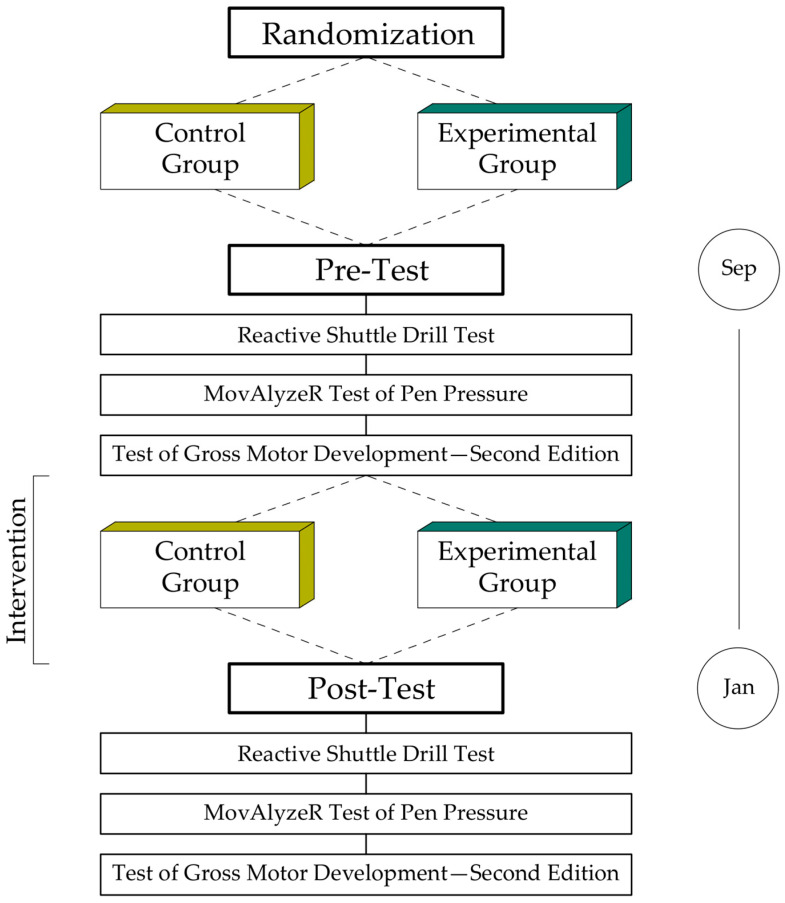
The experimental workflow. Classes were randomly assigned into control (no-Eduball) and experimental (Eduball) groups. Before and after the one-semester intervention, participants were tested in space-time orientation using the Reactive Shuttle Drill test, in graphomotor dexterities through the MovAlyzeR pen pressure test, and in fundamental movement (locomotor and object control) skills via the Test of Gross Motor Development (edition #2). The order of the tests was identical for each student and at each stage of the experiment.

**Figure 3 ijerph-19-15430-f003:**
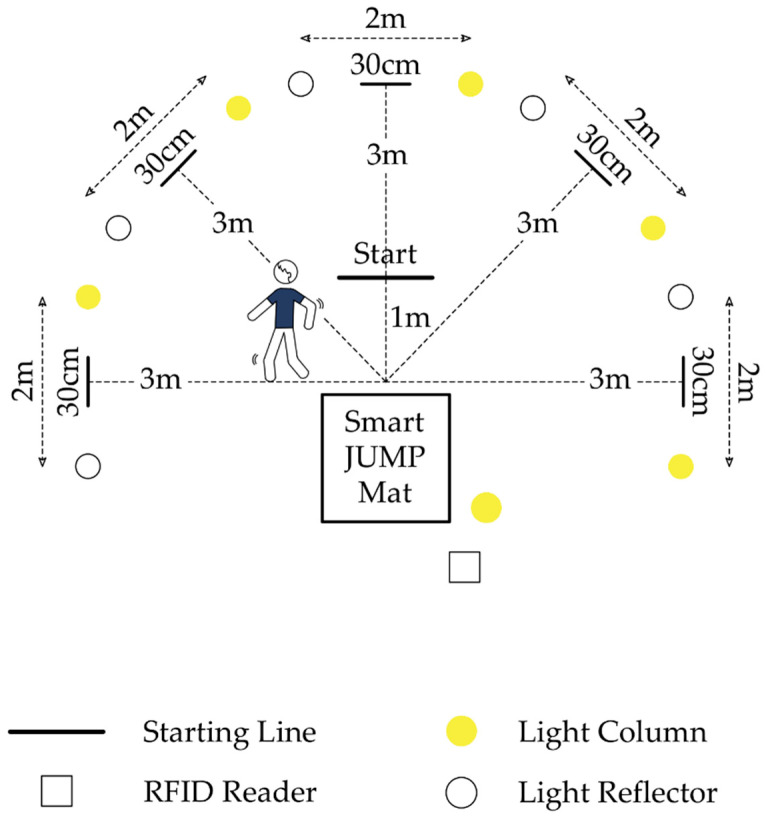
Reactive Shuttle Drill test. At the starting signal, the participant (standing back to the gates) began to run from the starting line to the mat, next to the randomly indicated (through the light) gate, and finally back to the mat (repeating the mat-gate-mat path five consecutive times). Each time, the participant had to (with both feet) cross the tape-marked line in the targeted gate and stand on the mat. The aim was to complete the test as quickly as possible.

**Figure 4 ijerph-19-15430-f004:**
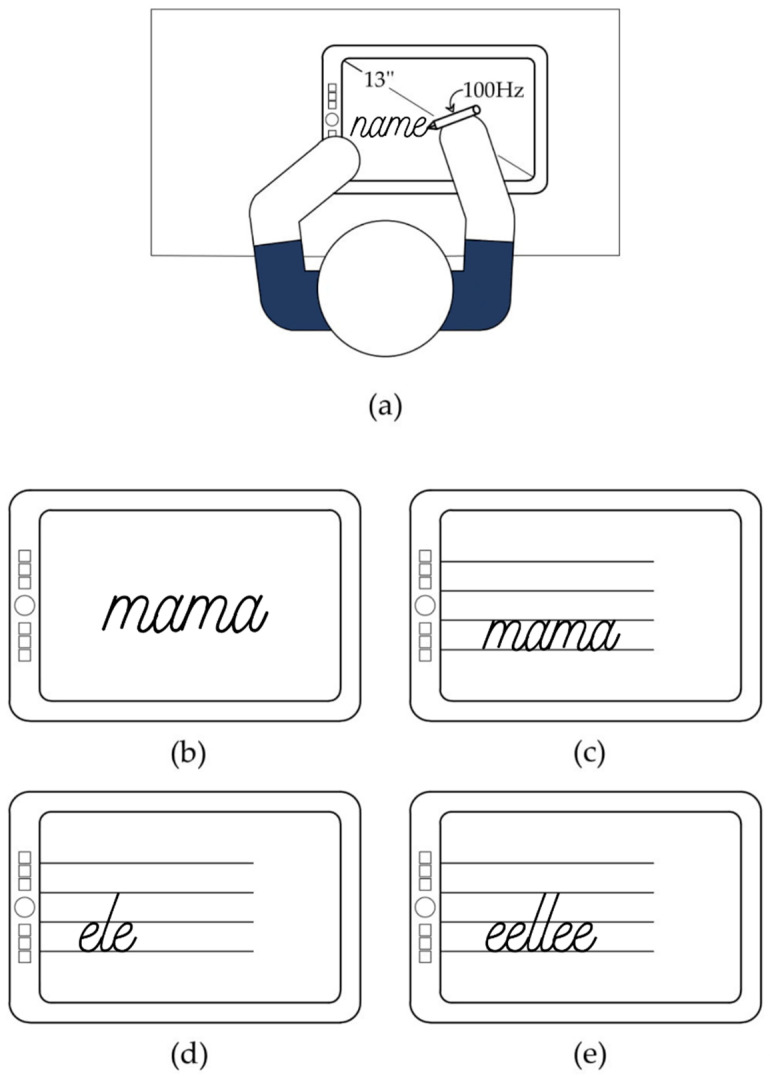
MovAlyzeR test of pen pressure. (**a**) Each participant took the test on a horizontally positioned graphics tablet with an electronic pen measuring pressure. Firstly, a student wrote his or her name twice as training (**b**–**e**); then, there were eight test trials consisting of performing four tasks twice: writing the word “mama” (the Polish equivalent of the English word “mom”) on a blank screen, writing the word “mama” in lines, writing the pseudo-word “ele” in lines, and writing the pseudo-word “eellee” (also in lines). During the whole test, participants used only their dominant hand.

**Figure 5 ijerph-19-15430-f005:**
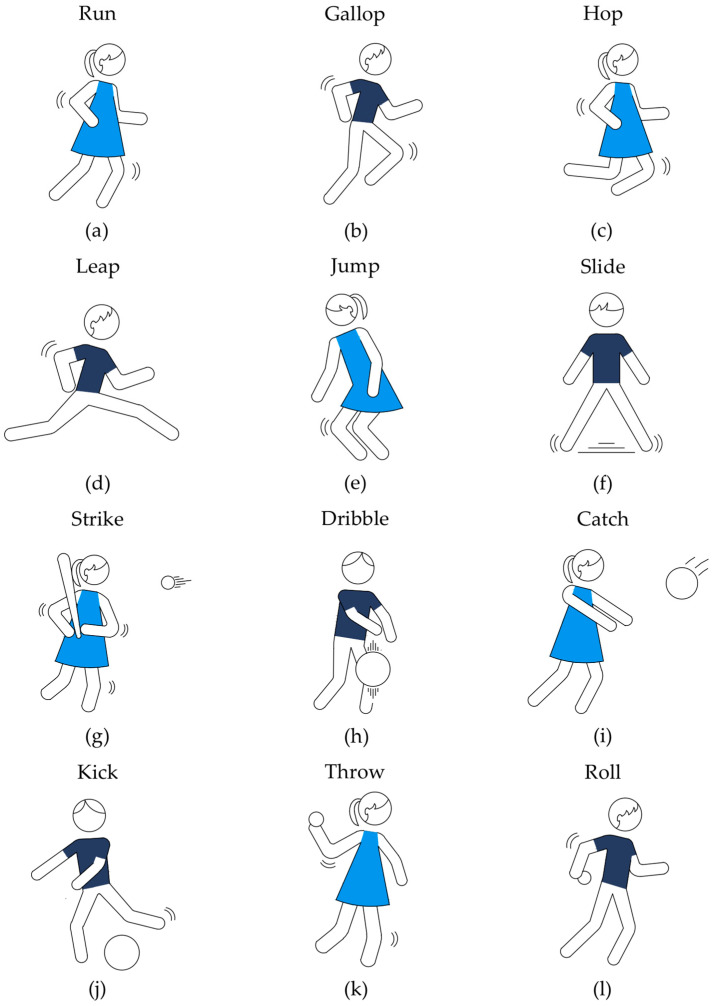
Test of Gross Motor Development—Second Edition. The test consisted of two subtests, each with six diverse aspects of gross motor development: (**a**–**f**) includes tasks in the locomotor subtest, and (**g**–**l**) includes tasks in the object control subtest. Three trials were conducted for each of the 12 items—one practice and two assessed.

**Figure 6 ijerph-19-15430-f006:**
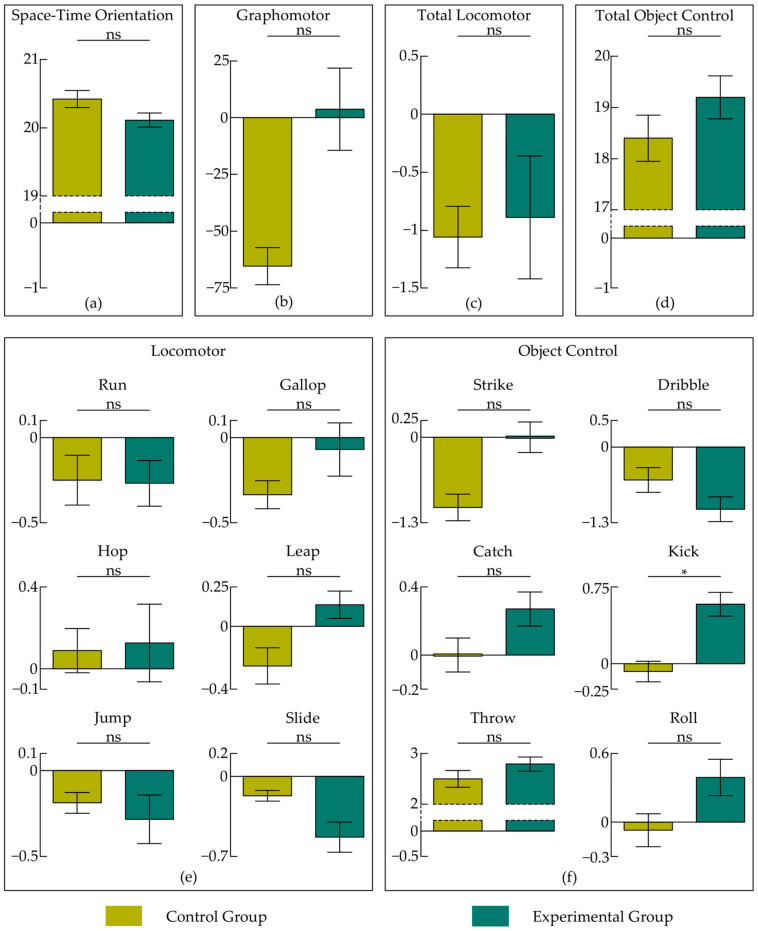
Motor progress in the control and experimental group was considered as changes (Δ) between post-test and pre-test (baseline) scores or—if necessary—as estimated post-test scores. (**a**) Estimated differences between groups’ post-test scores in space-time orientation; (**b**) differences between groups’ Δ in graphomotor skills; (**c**) differences between groups’ Δ in total locomotor skills; (**d**) estimated differences between groups’ post-test scores in total object control skills; (**e**) differences between groups’ Δ in particular locomotor skills, i.e., in running, galloping, hopping, leaping, jumping and sliding; (**f**) differences between groups’ Δ in particular object control skills, i.e., in striking, dribbling, catching, kicking and rolling, and estimated differences between groups’ post-test scores in throwing. Asterisk (*) indicates significant *p*-values (*p* < 0.05), and “ns” is not significant. Error bars depict standard errors of the means.

**Table 1 ijerph-19-15430-t001:** Mean and standard deviation of the control and experimental group in the pre-test and post-tests, and the paired samples *t*-test results.

Skills	Control Group							
	Pre-Test		Post-Test		Mean Difference	Student’s *t*	*p*	Cohen’s *d*
	Mean	*SD*	Mean	*SD*				
Space-Time Orientation	19.65	1.17	19.86	1.25	−0.21	−1.08	0.305	−0.31
Graphomotor	447.57	46.69	382.69	62.69	64.87	3.78	0.003	1.09
Locomotor	23.00	1.13	21.92	1.38	1.08	2.00	0.071	0.58
*Run*	3.75	0.45	3.50	0.91	0.25	0.82	0.429	0.24
*Gallop*	3.75	0.45	3.42	0.52	0.33	1.77	0.104	0.51
*Hop*	4.67	0.65	4.75	0.62	−0.08	−0.36	0.723	−0.11
*Leap*	2.92	0.29	2.67	0.65	0.25	1.15	0.275	0.33
*Jump*	4.00	0.00	3.83	0.39	0.17	1.48	0.166	0.43
*Slide*	3.92	0.29	3.75	0.45	0.17	1.48	0.166	0.43
Object Control	22.17	2.13	19.58	2.88	2.58	3.30	0.007	0.95
*Strike*	4.58	0.67	3.50	1.00	1.08	3.03	0.012	0.87
*Dribble*	3.92	0.29	3.33	0.99	0.58	1.86	0.089	0.54
*Catch*	2.75	0.62	2.75	0.62	0.00	0.00	1.000	0.00
*Kick*	3.75	0.45	3.67	0.49	0.08	0.43	0.674	0.12
*Throw*	3.58	0.79	2.83	1.27	0.75	1.83	0.095	0.53
*Roll*	3.58	0.67	3.50	0.67	0.08	0.29	0.777	0.08
**Skills**	**Experimental Group**							
	**Pre**-**Test**		**Post**-**Test**		**Mean** **Difference**	**Student’s** ** *t* **	** *p* **	**Cohen’s** ** *d* **
	**Mean**	** *SD* **	**Mean**	** *SD* **				
Space-Time Orientation	21.17	1.81	20.55	1.34	0.62	2.46	0.028	0.64
Graphomotor	360.33	94.92	363.68	87.14	−3.35	−0.09	0.927	−0.02
Locomotor	22.00	2.90	21.13	1.89	0.87	0.84	0.415	0.22
*Run*	3.80	0.56	3.53	0.74	0.27	1.00	0.334	0.26
*Gallop*	3.60	0.74	3.53	0.64	0.07	0.22	0.827	0.06
*Hop*	4.33	1.11	4.47	0.64	−0.13	−0.34	0.737	−0.09
*Leap*	2.73	0.46	2.87	0.35	−0.13	−0.81	0.433	−0.21
*Jump*	3.73	0.59	3.47	0.74	0.27	1.00	0.334	0.26
*Slide*	3.80	0.41	3.27	0.80	0.53	2.09	0.056	0.54
Object Control	18.13	3.38	18.33	3.22	−0.20	−0.23	0.818	−0.06
*Strike*	3.80	1.27	3.80	1.42	0.00	0.00	1.000	0.00
*Dribble*	3.27	0.88	2.13	1.25	1.13	3.90	0.002	1.01
*Catch*	2.60	0.63	2.87	0.35	−0.27	−1.29	0.217	−0.33
*Kick*	2.93	0.96	3.60	0.74	−0.67	−2.47	0.027	−0.64
*Throw*	2.53	1.13	2.53	1.30	0.00	0.00	1.000	0.00
*Roll*	3.00	0.93	3.40	0.74	−0.40	−1.31	0.212	−0.34

*SD*—standard deviation. Mean differences were calculated by the formula: pre-test scores—post-test scores.

**Table 2 ijerph-19-15430-t002:** Changes between post-test and baseline (pre-test) scores of the control and experimental group and the independent samples *t*-test results.

Skills	Δ		Mean Difference	Welch’s *t*	*p*	Cohen’s *d*
	Control Group	Experimental Group				
Space-Time Orientation	0.21	−0.62	0.83	2.60	0.016	0.99
Graphomotor	−64.87	3.35	−68.23	−1.72	0.101	−0.64
Locomotor	−1.08	−0.87	−0.22	−0.19	0.854	−0.07
*Run*	−0.25	−0.27	0.02	0.04	0.968	0.02
*Gallop*	−0.33	−0.07	−0.27	−0.75	0.459	−0.28
*Hop*	0.08	0.13	−0.05	−0.11	0.913	−0.04
*Leap*	−0.25	0.13	−0.38	−1.40	0.175	−0.55
*Jump*	−0.17	−0.27	0.10	0.35	0.734	0.13
*Slide*	−0.17	−0.53	0.37	1.31	0.205	0.49
Object Control	−2.58	0.20	−2.78	−2.41	0.024	−0.92
*Strike*	−1.08	0.00	−1.08	−1.86	0.074	−0.71
*Dribble*	−0.58	−1.13	0.55	1.29	0.210	0.50
*Catch*	0.00	0.27	−0.27	−0.90	0.377	−0.35
*Kick*	−0.08	0.67	−0.75	−2.26	0.033	−0.85
*Throw*	−0.75	0.00	−0.75	−1.52	0.145	−0.60
*Roll*	−0.08	0.40	−0.48	−1.15	0.260	−0.44

Δ—Changes/Delta calculated by the formula: post-test scores–pre-test scores. Mean differences were calculated by the formula: control group Δ—experimental group Δ.

**Table 3 ijerph-19-15430-t003:** The estimated difference in post-test scores between the control and experimental group and the analysis of covariance (*ANCOVA*) results (the result of the pre-test was set as the covariate).

Skills		*EMM* Difference	Sum of Squares	Mean Square	*F*	*p*	*η*²*p*
Space-Time Orientation	Covariate (Pre-Test)	−0.30	29.46	29.46	55.05	<0.001	0.70
	Group		0.73	0.73	1.35	0.256	0.05
Graphomotor	Covariate (Pre-Test)	−21.00	95.80	95.80	0.02	0.902	0.00
	Group		2241.90	2241.90	0.36	0.554	0.02
Locomotor	Covariate (Pre-Test)	−1.00	6.82	6.82	2.56	0.122	0.10
	Group		6.48	6.48	2.44	0.132	0.09
*Run*	Covariate (Pre-Test)	0.05	0.54	0.54	0.80	0.379	0.03
	Group		0.02	0.02	0.02	0.882	0.00
*Gallop*	Covariate (Pre-Test)	0.08	0.66	0.66	1.98	0.172	0.08
	Group		0.04	0.04	0.12	0.733	0.01
*Hop*	Covariate (Pre-Test)	−0.34	0.51	0.51	1.28	0.269	0.05
	Group		0.72	0.72	1.82	0.190	0.07
*Leap*	Covariate (Pre-Test)	0.16	0.20	0.20	0.76	0.394	0.03
	Group		0.16	0.16	0.61	0.441	0.03
*Jump*	Covariate (Pre-Test)	−0.43	0.26	0.26	0.68	0.416	0.03
	Group		1.11	1.11	2.93	0.100	0.11
*Slide*	Covariate (Pre-Test)	−0.50	0.06	0.06	0.13	0.720	0.01
	Group		1.62	1.62	3.49	0.074	0.13
Object Control	Covariate (Pre-Test)	0.80	53.83	53.83	7.08	0.014	0.23
	Group		2.78	2.78	0.37	0.551	0.02
*Strike*	Covariate (Pre-Test)	0.39	0.31	0.31	0.19	0.668	0.01
	Group		0.85	0.85	0.52	0.477	0.02
*Dribble*	Covariate (Pre-Test)	−0.83	3.90	3.90	3.29	0.082	0.12
	Group		3.68	3.68	3.10	0.091	0.11
*Catch*	Covariate (Pre-Test)	0.12	0.02	0.02	0.08	0.776	0.00
	Group		0.10	0.10	0.40	0.531	0.02
*Kick*	Covariate (Pre-Test)	0.07	0.45	0.45	1.09	0.307	0.04
	Group		0.03	0.03	0.07	0.797	0.00
*Throw*	Covariate (Pre-Test)	0.30	7.84	7.84	5.61	0.026	0.19
	Group		0.44	0.44	0.31	0.581	0.01
*Roll*	Covariate (Pre-Test)	−0.12	0.01	0.01	0.03	0.868	0.00
	Group		0.08	0.08	0.15	0.698	0.01

*EMM*—estimated marginal mean. *EEM* differences were calculated by the formula: estimated post-test scores of the control group—estimated post-test scores of the experimental group. *F*—*ANCOVA F*-test value.

## Data Availability

The data presented in this study are openly available in the Open Science Framework at https://osf.io/6pfrt (accessed on 8 September 2022).

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
