# Peer review of "No Motor Costs of Physical Education with Eduball"

_ijerph, 2022, doi:10.3390/ijerph192315430_

Round 1

Reviewer 1 Report

Congratulations on taking on such an interesting topic.

I have a few questions and suggestions:

- The experiment involved 7-8 year old students (first semester of the school year). Does this mean that the experiment was started by children who just started the school year for the first time? Did you randomize the children into groups and include them in the study without knowing whether they had contraindications to PE, and whether they regularly attended classes? When was the procedure for drawing the children? Could this mean that the intervention did not last the full 5 months?

Wasn't it better to do the study in the 2nd semester of the school year as the children are already familiar with the school and physical education classes, or to run it longer, e.g., the whole year as highlighted in the limits? 

- Why were Eduball classes conducted only 2 and not 1 or 3 times a week in the experimental class? Why such a decision, the Authors did not explain it? It can be supplemented in the text.

- Was children's out-of-school physical activity considered in the inclusion or exclusion criteria as confounding variables? 

- Shouldn't the results of the experiment with regard to motor tests be performed with controls for the development of children's somatic characteristics? 

- Did the children in the control group exercise with their entire class, and the children in the experimental group only in a drawn group?

Author Response

Dear Reviewer,

Sincerely,
Ireneusz Cichy
& Michal Klichowski

Reviewer 2 Report

Dear editor and authors,

Thank you for the opurtunity to read this very interesting work and offer my advice as to how it could be improved. 

The paper is relevant to the submitted outlet. 

The paper is a replication which I always consider good (Eduball experiments).

The paper is a replication with a "twist" (motor vs cognitive focus) which is very good.

The paper offers novel results which is excellent (no motor cost during the motor-oriented Eduball intervention).

These said my impression is that the first thing the paper needs is a careful reconsideration of the syntax and the phrasing throughout. I am writing this with the best of intentions and being myself a non-native English speaker. The paper has many odd choices for phrasing arguments, many very long (nigh Ciceronian) sentences and many quite unclear sequences in its local structure. The authors could benefit from:

1. Finding the exact words for what they would like to phrase.

2. Making - in some occasions - smaller sentences.

3, Checking whether subsequent sentences make sence. 

I would dwell more on the subject but - to their credit - the authors were able to communicate the concepts they wanted to and included a very good level introduction and discussion. Therefore, none of the above is catastrophic. It is definetely awkward to read though and I personally believe they can do better based on my perception of their skils. 

I invite the authors to consider if Figure 1 really belongs to the introduction? I will accept a reasonable defence of their choice.

My sincere apologies but I am not entirely sure how 27 participants can provide a P(1-β) = .8 and I am almost enirely sure that 26 participants cannot possibly provide P(1-β) = .99.  Running their data parameters myself in GPOWER I get completely different results. I might be doing something wrong. Could the authors provide me with a screenshot of their GPOWER output that confirms theirs calculations?

The next few paragraphs after the power calaulations include so many information in brackets that I would like to invite the authors to consider whether they would be able to confer their message in a less fragmented way? 

The methodology is sound and portrayed well in figures although the "text brackets" make it hard to follow without the visuals. The analyses could benefit from the exclusion - or at-least the minimization - of the discourse of basic statistical terms in lines 275-293. I feel most readers will be acquainted with these. Again here if the authors would like to offer a reasonable defence of their choice I will accept it.

I like the novelty of the results but I believe that at-least Figure 6 should be in text. It was very hard to follow: the lines converge and overlay on each other, the bars are confusing and the overall result is hard to understand. The outcome is not at the same level with the rest of the paper. I invite the authors to consider whether text could be a more appropriate way of informing their audience.

I have some conceptual considerations as to how the results came about that I prefer to discuss in a possible revision. I am sorry to have given you an already hard homework for your hard and meticulous work and I hope that you will be able to overcome these difficulties and improve the paper further. 

It was very interesting reading your work. 

All my best,

Author Response

(The authors gave the same response as above.)

Round 2

Reviewer 2 Report

Dear editor and authors,

       Thank you kindly once more for the oppurtunity to re-visit and enjoy reading this fine manuscript, 

The authors have made a Herculian effort to address my comments as best they could in a rather impressive (record(?)) author response time. Their hard work is truly commendable. 

I like the paper very much which I did not previously state on purpose, firstly, because it is not about what I Iike - it's about science - and secondly because it sometimes makes people less inclined to work hard; clearly it seems this would not have been the case here.

Having admitted to the fact that I like the paper very much, all my concerns have been resolved. There is room for improvement (where is there not?) but I can safely say that my personal opinion is that the scientific value of the paper merits publication as is.

I would only like to note that I assume the authors submitted the manuscript in word format. It is downloadable in PDF format. I had to convert and accept changes to understand the differences. Naturally, the conversion process is never perfect and I was a bit sad not to have seen the paper in the format that it will likely be published.

Perhaps the editor feels the same? and a version with "accept changes" is in order? or maybe the "track record" version was specifically addressed to me and the editor has the final ms? (I would not be surpised given the amount of effort the authors have already dedicated here if that were the case).

That is all from me. I will not dwell - as I intended - to the conceptual issues that concerned me relating to the results, The clearness of the text has allowed me to see past them.

Congratulations for composing a fine manuscript with novel and contributing findings.